# Intramuscular Hematomas in Patients Receiving Prophylaxis or Anticoagulant Treatment after Spinal Cord Injury (SCI)—A Rare Complication: Description of Seven Cases and a Literature Analysis

**DOI:** 10.3390/biomedicines11041142

**Published:** 2023-04-10

**Authors:** Magdalena Mackiewicz-Milewska, Małgorzata Cisowska-Adamiak, Iwona Szymkuć-Bukowska, Katarzyna Sakwińska, Iwona Domarecka, Anna Lewandowska, Iwona Głowacka-Mrotek

**Affiliations:** Department of Rehabilitation, Nicolaus Copernicus University in Toruń, Collegium Medicum in Bydgoszcz, 85-094 Bydgoszcz, Poland

**Keywords:** spinal cord injury, anticoagulation, intramuscular hematoma

## Abstract

Spinal cord injuries (SCIs) are associated with a high risk of thromboembolic complications (VTE), despite the use of antithrombotic prophylaxis in the form of low-molecular-weight heparin (LMWH). The occurrence of VTE requires, as in other diseases, full-dose antithrombotic treatment. Herein, we describe seven cases of soft tissue hemorrhagic complications in the form of spontaneous intramuscular hematomas (SMHs) in patients after SCI undergoing rehabilitation. Four patients received anticoagulant therapy due to previously diagnosed deep vein thrombosis (DVT), and three patients received anticoagulant prophylaxis. None of the patients had a significant injury immediately before the hematoma appeared, and the only symptom was a sudden swelling of the limb without accompanying pain. The hematomas in all patients were treated conservatively. In three patients, significant decreases in hemoglobin were observed; in one patient, a blood transfusion was required for this reason. In all patients treated via anticoagulation, the anticoagulation treatment was modified at the time of diagnosis of the hematoma; in three patients, oral anticoagulants were changed to LMWH in a therapeutic dose, while in one patient, anticoagulant treatment was completely discontinued. Conclusions: Intramuscular hematomas are rare complications after SCI. Each sudden swelling of a limb requires ultrasound-based diagnostics. At the time of diagnosis of a hematoma, the level of hemoglobin and the size of the hematoma should be monitored. The treatment or anticoagulation prophylaxis should be modified if necessary.

## 1. Introduction

SCIs (spinal cord injuries) are among the most devastating to the human body. Among numerous complications, there are also thromboembolic complications (VTE), which may be life-threatening for patients.

A high risk of VTE, including deep vein thrombosis (DVT) and pulmonary embolism (PE) after SCI, is associated with the presence of all elements of Virchov’s triad: venous stasis, hypercoagulability, and endothelial damage [1,2,3]. The incidence of DVT in these patients in the acute phase in the era of thromboprophylaxis is 4.8–27%; before the era of prophylaxis, it was 48–100% [4,5,6].

Due to the increased risk of VTE, thromboprophylaxis is common among patients with SCI. The time at which prophylaxis is started should be as early as possible, but after the risk of bleeding complications has decreased [2,7]. Despite the use of prophylaxis in the form of LMWH, complications of DVT and PE occur in this group of patients, especially in the early phase after the injury, i.e., up to three months, which then requires full-dose anticoagulant therapy [8]. Anticoagulant treatment is associated with the possibility of bleeding, sometimes even becoming life-threatening. Some studies have shown that 0.5–4% of patients receiving LMWH anticoagulant therapy experience major bleeding [9]. Herein, we present a series of seven patients diagnosed with spontaneous intramuscular hematomas (SMH) in a lower limb. All patients received either LMWH prophylaxis or full-dose anticoagulation therapy.

## 2. Materials and Methods

Observations were carried out in the years 2011–2022 in the Rehabilitation Department of the University Hospital No. 1 in Bydgoszcz. A retrospective analysis included 152 patients with traumatic (TSCI) and non-traumatic SCI (NTSCI) who were undergoing rehabilitation at different times after the injury. The study was approved by the Local Bioethics Committeeof the Nicolaus Copernicus University in Toruń, Collegium Medicum in Bydgoszcz (no. KB 45/2023). The study did not collected personal data of the participants and was completely anonymous. In our center, the standard procedure among this group of patients is the administration of LMWH in a prophylactic dose during early rehabilitation until being discharged home, as well as administration of antithrombotic prophylaxis in chronic patients during the so-called late rehabilitation period during their stay in the Rehabilitation Department. It is also standard to start a therapeutic dose of LMWH in patients diagnosed with DVT or PE during their stay, as well as deliver oral anticoagulant treatment in the later period. Herein, we present a short description of seven cases of patients who were diagnosed with SMH during their stay in the Rehabilitation Department, none of whom suffered a significant injury, e.g., a fall or a blow. After diagnosis of the hematoma, four patients received anticoagulation treatment and three patients received anticoagulant prophylaxis. The mean time from the injury to the onset to the occurrence of muscle hematoma was 10.4 weeks.

## 3. Series of Cases

### 3.1. Case No. 1

A patient with spastic paresis of the lower limbs, suffering from myelopathy in the course of an advanced degenerative disease of the vertebral discs in the thoracic spine. A month before admission, the patient was operated on at the Th6–Th8 level. In the physical examination, on admission, a bilaterally increased tone in the lower limbs was observed: 2 on the Modified AshworthScale (MAS), weakened muscle strength in the lower limbs, 2 on the Medical Research Council (MRC)Scale for Muscle Strength, disturbed surface sensation from the level of Th7, and lesion type B of the American Spinal Injury Association (ASIA) impairment scale. The patient was functionally wheelchair-bound and receiving a prophylactic dose of LMWH of 0.4 mg s.c. of enoxaparinum. One month after admission, the patient developed swelling of the lower right limb, without pain or redness. Upon ultrasound of the lower limbs, DVT was excluded, and hematomas were found in the muscles of the right thigh (semitendinous, sartorius), the dimensions of the larger one of which was 40 × 40 × 10 mm with partial damage to the tendons. The patient denied any injury during the previous days. The size of the hematomas was controlled by ultrasonography (US), which revealed gradual resorption. There was no decrease in the level of hemoglobin (Hb). Conservative treatment was carried out, and the doses of anticoagulation prophylaxis were not modified. The time from injury to SMH was 8 weeks.

### 3.2. Case No. 2

A patient was admitted due to plegias of the lower limbs after multi-organ injury a month earlier, as a consequence of a fall from a height with a spinal injury at the level of Th12 and L1–L4. Implantation of Th10–S1 stabilization with decompression of nerve structures took place on the day of injury. Upon admission to the Rehabilitation Department, examination of the lower limbs revealed reduced muscle tone and muscle strength in all muscle groups of the lower limbs: 0 on the MRC scale, reduced surface sensation from the level of Th11, diminished below L2, ASIA level A, and lower limbs without edema or pain. The patient received a prophylactic dose of LMWH of 0.4 mg s.c. of enoxaparinum. Due to the persistently high D-dimer value of 8035 mg/L, a Doppler US examination of the lower limbs was performed, revealing a massive proximal DVT bilaterally in the lower limbs; therefore, a treatment dose of heparin was implemented (enoxaparin 2 × 1 mg/kg b.w.s.c.). Two weeks later, without any obvious injury, the patient developed a painless edema of the right lower limb. Extensive diagnostics included ultrasound and an angio-CT (computed tomography), revealing a massive hematoma in the thigh and right iliac muscle (hematoma 120 × 80 × 50 mm in the rectus femoral insertion and a large median hematoma, with a hematoma in the head of the rectus femoris and in the right iliac muscle being 62 × 54 mm (Figure 1).

Due to visible active bleeding in the CT scan of the abdomen and pelvis with contrast, LMWH was discontinued after consultation with a vascular surgeon. There was also a decrease in Hb levels, requiring blood transfusions. A vascular filter was implanted into the inferior vena cava. In the following days, significant intensification of the edema of the lower limbs was found, and control Doppler US examination revealed intensification of the thrombotic changes in the lower limbs. A control angio-CT of the abdominal cavity and pelvis was performed, excluding active bleeding, and LMWH was started initially at half of the therapeutic dose of 2 × 0.5 mg/kg b.w., followed by a full dose. The size of the SMH was monitored via US examination and its reduction and resorption were observed. Hb levels were monitored, falling from 11.4 to 8.1 g/dL with hemodynamic instability, resulting in the patient requiring a blood transfusion. The time from injury to SMH was 6 weeks.

### 3.3. Case No. 3

A patient, admitted due to paraplegia after a traffic accident, with a fracture of the spine at the Th1–Th2 level. The injury occurred two months before admission. As a result of the injury, she also suffered multi-place fractures of the ribs and long bones of the lower and upper limbs. The physical examination revealed paralysis of the lower limbs: muscle strength 0 on the MRC scale, loss of superficial sensation from the level of Th3, and reduced muscle tone. Functionally, the patient had to stay in bed and had not adapted to a wheelchair. On the ASIA scale, the patient had damage type A. On the fifth day of hospitalization, proximal DVT of the left lower limb was diagnosed (the examination was performed due to high d-dimer values). Initially, a therapeutic dose of enoxaparin was included in the treatment, and then, after three weeks, dabigatran was introduced at a dose of 2 × 150 mg/os. During anticoagulant treatment, on Day 7 of dabigatran treatment, a large hematoma was observed on the inside of the left lower leg and thigh on the anterior, anteromedial, and anterolateral sides, with a massive edema of the limb without pain. US examination revealed several SMHs within the right thigh; in the front, upper third of the thigh, it measured 72 × 40 × 70 mm; in the anteromedial upper third of the thigh, it measured 62 × 28 × 81mm. Dabigatran was discontinued and low-molecular-weight heparin was started at a therapeutic dose of 2 × 1 mg/kg b.w.s.c. The size of the hematomas was monitored under US examination. We observed their reduction and partial resorption. The patient was consulted surgically and did not require surgical decompression of the hematomas. Hb levels were monitored, and no significant decrease was observed. The time from injury to SMH was 14 weeks.

### 3.4. Case No. 4

A patient was admitted due to tetraplegia. Spinal injury at the cervical level—an explosive fracture of the C7 vertebral body—occurred as a result of a jump into shallow water a month before admission to the department. Since the injury, he had been taking a prophylactic dose of LMWH enoxaparin of 0.4 mg s.c. On admission, the physical examination revealed paresis of the upper limbs, a hand muscle strength of 3 on the MRC scale, muscle strength of the lower limbs of 0 on the MRC scale, muscle tone in the lower limbs of 2 on the MAS, sensory disturbances from the C7 level, and ASIA type A lesions. Functionally, upon admission, the patient had adapted to a wheelchair. On the 37th day of hospitalization, edema of the left lower limb was present, mainly in the shin and ankle, with a difference in circumference and increased warmth of the limb. The injury was negated; however, we observed the patient while moving from the bed to the wheelchair, and the limbs hit the wheelchair frame. US examination revealed a hematoma in the muscles of the posterior group of the left shin and in the bicep femoris muscle of the left thigh, 55 × 30 × 120 mm in size, and DVT was excluded. A CT scan of the lower left limb showed a gastrocnemius hematoma of 48 × 23 mm in size and heterotopic ossification (HO) around both hip joints. The patient was monitored with subsequent US examinations, and we observed resorption of the hematoma. The level of hemoglobin was also monitored, with no significant decrease observed. A prophylactic dose of LMWH was therefore not administered. The time from injury to SMH was 9.4 weeks.

### 3.5. Case No. 5

A patient was admitted, due to lower limb paralysis caused by an intramedullary tumor (Schwannoma) at the height of Th10, one month after surgery. The physical examination on admission revealed flaccid paralysis of the lower limbs, loss of superficial sensation from the level of Th10, and edema in both lower legs. A Doppler US examination of both lower limbs was performed, which revealed features of DVT of the distal veins of both legs. The treatment included a therapeutic dose of LMWH of 2 × 1 mg.kg b.w., and then the oral anticoagulant warfarin after six weeks. After 10 days of treatment with warfarin (INR 2.2), massive swelling of the right thigh was observed, and ultrasound examination was performed again. There were no new thrombotic changes in the deep veins of the thigh and right shin; however, three hematomas were found in the posterior group of thigh muscles—48 × 39 × 10 mm, 53 × 47 mm, and 35 × 39 mm in size, respectively (Table 1). Due to a history of DVT, it was decided to continue anticoagulation, but oral drugs were changed to LMWH at a dose of 2 × 1 mg/kg b.w. The patient denied an injury. In subsequent US examinations, the process of absorption of the hematomas was monitored, and reduction of the hematomas was observed during the ultrasound. The level of Hb was also monitored, and a decrease in the level from 11.2 to 8.8 g/dL was observed. The patient was hemodynamically stable, and thus did not require a blood transfusion. The time from injury to SMH was 11.4 weeks.

### 3.6. Case No. 6

A patient was admitted to the department due to severe paresis of the upper limbs and paralysis of the lower limbs three months after injury. Injury of the cervical spine was present—with C5, C6, and C7 fractures and subluxations in the facet joints C4–5 and C5–C6. During the examination upon admission, the patient had muscle strength on the MRC scale of 0 in the lower limbs, while in the upper limbs (wrist flexors and extensors), they scored 3 on the MRC scale. The muscle tone in the upper and lower limbs increased to 1 on the MAS scale. Weakened superficial sensation to the level of C7, lifted from the level of Th3, and type A on the ASIA scale were observed. The patient had functionally adapted to a wheelchair. Due to previously diagnosed DVT of the proximal veins of both legs, rivaroxaban at a dose of 20 mg p.os was administered. On the 16th day of hospitalization, swelling of the left thigh was observed, without pain. He denied injury. US examination of the left lower limb showed an SMH in the posterior group of the left thigh muscles along almost the entire length, with transverse dimensions of 60 × 40 mm, with a bloody subcutaneous adipose tissue with small streaks of fluid. Rivaroxaban (Xarelto) was discontinued and LMWH was introduced at 1 × 1 mg/kg b.w. The size of the hematoma was monitored under US examination. Hb level also was monitored, and a decrease from 12.9 to 9.4 g/dL was observed, meaning that he did not require a blood transfusion. The time from injury to SMH was 14 weeks.

### 3.7. Case No. 7

A patient was admitted to the department for 2.5 months after a traffic accident, which resulted in multiple traumas, including a craniocerebral trauma, a C7 spinal injury, a fracture of the Th4 vertebra with a large hematoma at the level of the broken body, a fracture of the Th5 body, a fracture of the transverse processes of the Th2 and Th3 vertebrae, a fracture of at L1–L3, a fracture of the shaft and a transtrochanteric fracture of the right femur, and fractures of the shafts of both bones of the right lower leg. On admission, examination revealed paresis of the upper limbs in the hand, with a muscle strength of 2 on the MRC scale and a muscle strength in the lower limbs of 2 on the MRC scale. However, muscle tone was difficult to assess due to fixed joint contractures. Functionally, the patient was confined to lying and was not wheelchair-adapted. The patient received a prophylactic dose of LMWH. On the first day of hospitalization for rehabilitation, swelling of the lower leg and right thigh occurred. US examination in the medial part of the right thigh showed edema of the skin and subcutaneous tissue, an intramuscular hematoma with hyperechoic foci (length of 90 × 122 × 40 mm), and features of periarticular ossification of the right hip joint confirmed via X-ray. The size of the hematoma was monitored by ultrasound, demonstrating its resorption, and the level of Hb was monitored, with no significant decrease observed. The time from injury to SMH was 10 weeks.

All cases are summarized in Table 1. A list of all antithrombotic treatment regimens implemented until SMH formation is provided in Table 2.

## 4. Discussion

All the patients presented by us who were diagnosed with SMH in the lower limbs had antithrombotic prophylaxis (*n* = 3) or full antithrombotic treatment (*n* = 4). The use of anticoagulant prophylaxis in patients with acute SCI is recommended in many countries, including [10]. According to most sources, the duration of antithrombotic prophylaxis is two to three months after the injury or until leaving the Early Rehabilitation Unit. The length depends on whether the SCI is complete or not and on the presence of risk factors such as obesity, heart failure, fractures of the limbs and pelvis, heart failure, ch. cancer, aged over 70 years, and history of VTE [3,5,11]. The Consortium for Spinal Cord Medicine [3] and Geerts [12] put forward the possibility of using oral anticoagulation in patients after SCI during rehabilitation to prevent VTE.

Spontaneous muscle hematomas (SMHs) as a complication of anticoagulation or antithrombotic prophylaxis are rarely described in the literature. It has been reported that the frequency of SMH activation in patients undergoing anticoagulation is 0.6% [13]. These hematomas mainly affect the abdominal muscles, and may spread to the peritoneal or retroperitoneal space or may be contained by the fascia [14]. The most commonly affected area is the rectus sheath [15,16,17,18]. Hematomas within the lumbar and iliac muscles have also been described, which may lead to hemodynamic instability and could potentially be fatal [19,20]. SMHs can be divided into small ones, which are clinically insignificant and most often remain contained in the fascia—mainly treated conservatively—and large ones that can cause hemodynamic instability and may be life threatening. From the literature analysis, it appears that large SMHs can be surgically evacuated by incision and drainage of the hematoma or via surgical excision and drainage [21]. Another method is embolization of the bleeding vessel (percutaneous transcatheter arterial embolization, PTAE) in the case of active bleeding diagnosed by computed tomography (CT) [22].

Active bleeding is most commonly seen, requiring embolization or drainage and a blood transfusion [13].

The most often described risk factors for the development of SMHs are microtrauma, coagulation disorders (requiring anticoagulant treatment), chronic renal insufficiency and hemodialysis, hypertension, cardiac and hepatic insufficiency, closed glottis straining, degenerative muscle diseases, and congenital collagen diseases [13,16,21].

In the majority of our patients, there were no other risk factors for spontaneous bleeding other than anticoagulation or antithrombotic prophylaxis. One patient (case no. 2) had hypertension, which is listed as a risk factor for SMH. In addition, it can be assumed that intensive rehabilitation could lead to micro-injuries of the muscles. We found no other bleeding complications in any of the patients in this case series. All described patients had normal liver and kidney parameters, which were monitored several times during hospitalization.

The main symptoms of SMH are pain at the site of SMH formation, emphasized by numerous authors, and, depending on the location and size of the swelling in a given area, a decrease in hemoglobin and hemodynamic instability [18,19,20]. Intramuscular hematomas in patients after SCI are rarely reported. To the best of our knowledge, this is the largest group of patients diagnosed with intramuscular hematomas. Difficulties in diagnosis are caused mainly by a lack of pain in this group of patients, caused by pain sensation disorders after SCI. In our group of patients, the main complaint was sudden swelling of the limb, sometimes with increased warmth and a decrease in Hb level, as observed in three of our patients, one of whom required a blood transfusion. In the available literature, we found four reports of SMH in patients after SCI. Carpentier [23] described three cases of spontaneous muscle damage in patients with SCI—all patients had SMH of the lower limb, two musculus semimembranosus and one adductor muscle. These patients, similarly to ours, had swelling of the limb, with only one person feeling pain (ASIA C). No significant injury was found and a diagnosis was made, as in most of our patients, on the basis of ultrasound. All patients described by Carpentier [23] had intense spasticity, which the authors associated with the cause of muscle damage during rehabilitation—muscle stretching and transfers. None of the described patients received anticoagulant treatment or prophylaxis, which differed from the patients described by us.de Almeida [24] also described two cases after SCI at the thoracic level, in the chronic phase nine months after the injury and four months after that. They were diagnosed with large gluteal hematomas that required surgical treatment. Both patients took LMWH prophylactically, which was implemented after admission to the Rehabilitation Department. Neither patient had an injury, and both had edema of the hip area as the main symptom. In both cases, at the beginning of rehabilitation enoxaparine (40 mg once daily) LMWH prophylaxis was introduced. Hematomas occurred in the third and first weeks of prophylaxis, respectively. Yeung [25], in his work, described three cases of massive intramuscular hematomas in the lower limbs in patients after spinal cord injury due to various reasons: hematoma (spinal subdural hematoma), trauma, and ischemia of the spinal cord. All patients were fully anticoagulated with i.v. heparin or enoxaparin for various reasons. In two cases, the hematomas required incision and drainage, while in all patients, the dose of heparin was reduced to prophylactic—and implantation of the inferior vena cava filter occurred in one case. Another paper describing hematomas in patients after SCI was presented by Snoecx [26]. He described four cases of patients with suspected HO, in whom, prior to making the diagnosis of HO, a hypoechoic space was diagnosed in the iliac muscle suggestive of a hematoma with tearing of the psoas muscle. All described patients had spasticity, and three out of the four were taking anticoagulants. The author suggested that hematomas, i.e., muscle microtrauma, may precede the formation of HO. The cases presented in the literature and our own observations show that SMH in patients after SCI can occur both in the early and chronic periods after the injury, and can be associated with the use of anticoagulant prophylaxis or anticoagulant treatment and with microtrauma caused during rehabilitation, especially in patients with spasticity.

In our group, five out of the seven presented patients had increased muscle tone on the MAS scale, while the patient with the largest hematoma requiring a blood transfusion had flaccid tone in the lower limbs. Both stretching exercises commonly used in this group of patients and learning transfers from the bed to a wheelchair may be associated with microtrauma, which may eventually lead to the formation of intramuscular hematomas.

In the cases described by us, the cause of the hematomas seems to be mixed. On the one hand, the use of drugs or antithrombotic prophylaxis could have been the cause, while on the other hand, wet injuries are likely the reason. The clinical symptoms of SMH in this group of patients are extremely poor, as the basic symptom of pain is not observed and only edema of the limb is observed. Ultrasound remains the basic diagnostic tool, with CT in cases of large hematomas with a decrease in hemoglobin. It is also very important to monitor the level of Hb at the same time, which will aid in detecting active bleeding and help in making decisions about discontinuing anticoagulant drugs.

## 5. Conclusions

Observation of swelling of the lower limb in a patient after SCI should be considered in the differential diagnosis of SMH. Quick diagnosis and appropriate monitoring of the patient (ultrasound and hemoglobin checks) allow for adequate therapy and make it possible to make decisions about discontinuing or modifying anticoagulant treatment and discontinuing or leaving anticoagulant prophylaxis. It should be remembered that discontinuation of thromboprophylaxis is always associated with a high risk of developing DVT, especially in patients with recent injury. Similarly, discontinuation of anticoagulant therapy in patients with SMH after SCI and previously diagnosed DVT may be associated with worsening of VTE, and SMH patients must be monitored for this as well, especially if decisions are made to discontinue LMWH or oral anticoagulants. In patients with SMH, it is necessary to monitor the size of the hematoma under US examination and with systematic control of Hb.

## Figures and Tables

**Figure 1 biomedicines-11-01142-f001:**
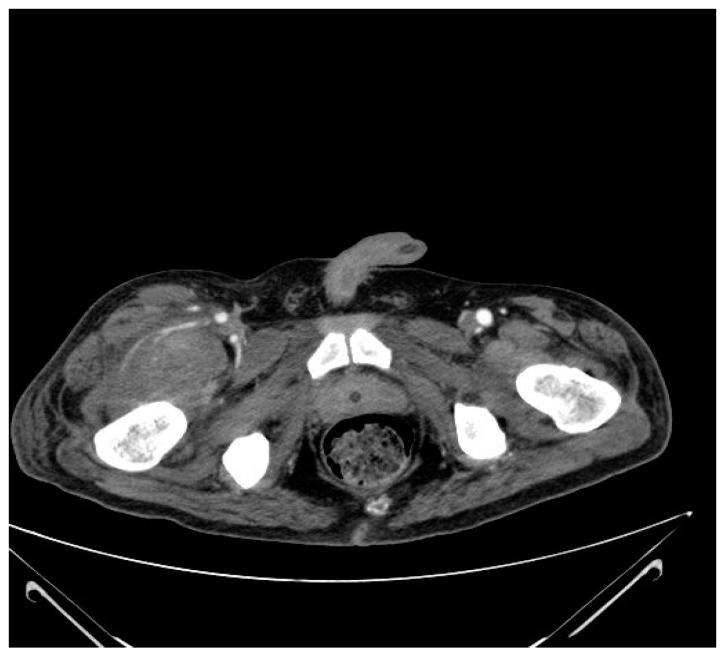
CT scan with contrast of the pelvis in the transverse plain showing the hematoma and active bleeding in the right iliac muscle.

**Table 1 biomedicines-11-01142-t001:** Characteristics of the cases.

Case No.	Type of SCI	Level of Injury	Muscle Tension ofthe Affected Limb (MAS)	LMWH Prophylaxis at the Time of SMH Appearance	Anticoagulation Therapy at the Time of SMH	Modification of Anticoagulation Therapy or Prophylaxis	SMH Symptoms	SMH Localization
1	NTSCI	Th6–Th8	2	Yes	No	No	Edema of the right lower limb	Semitendinosus and sartorius, right limb
2	TSCI	Th12 and L1–L4.	0	No	Yes	Yes	Edema of the right lower limb	Illiacus, right limb
3	TSCI	Th1–Th2	0	No	Yes	Yes	Edema of the right lower limb	Multifocal in tight, right limb
4	TSCI	C7	2	Yes	No	No	Edema of the left lower limb	Gastrocnemius, left limb
5	NTSCI	Th9–Th10	0	No	Yes	Yes	Edema of the right lower limb	Posterior thigh muscle group, right limb
6	TSCI	C4–C7	1	No	Yes	Yes	Edema of the left lower limb	Posterior thigh muscle group, left limb
7	TSCI	C7 and Th5	ND	No	No	No	Edema of the right lower limb	Medialis thigh muscle group, right limb

Legend: MAS, Modified AshworthScale; LMWH, low-molecular-weight heparin; ND, non-diagnostics; TSCI, traumatic spinal cord injury; NTSCI, non-traumatic spinal cord injury; SMH, spontaneous intramuscular hematoma.

**Table 2 biomedicines-11-01142-t002:** A list of all antithrombotic treatment regimens implemented in a given case in chronological order until SMH formation.

Case Number	Case 1	Case 2	Case 3	Case 4	Case 5	Case 6	Case 7
Type of therapy	(1) Enoxaparin (0.4 mg s.c.) for 8 wk	(1) Enoxaparin (0.4 mg s.c.) for 4 wk Followed by: (2) Enoxaparine (2 × 1 mg/kg b.w.) for 2 wk	(1) Enoxaparin (0.4 mg s.c.) for 9 wk Followed by: (2) Enoxaparine (2 × 1 mg/kg b.w.) for 3 wk (3) Dabigatran (2 × 150 mg) for 1 wk	(1) Enoxaparin (0.4 mg s.c.) for 9.4 wk	(1) Enoxaparin (0.4 mg s.c.) for 4 wk Followed by: (2) Enoxaparine (2 × 1 mg/kg b.w.) for 6 wk Followed by: (3) Warfin for 1.4 wk	(1) Enoxaparin (0.4 mg s.c.) for 7.7 wk Followed by: (2) Rivaroxaban (20 mg) for 6.3 wk	(1) Enoxaparin (0.4 mg s.c.) for 10 wk

Legend: wk, weeks; b.w., body weight; s.c., subcutaneous.

## Data Availability

The data presented in this study are available on request from the corresponding author. The data are not publicly available due to privacy.

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
