# Peer review of "Intramuscular Hematomas in Patients Receiving Prophylaxis or Anticoagulant Treatment after Spinal Cord Injury (SCI)—A Rare Complication: Description of Seven Cases and a Literature Analysis"

_biomedicines, 2023, doi:10.3390/biomedicines11041142_

Round 1

Reviewer 1 Report

The paper is interesting and within the scope of the journal. I only have some questions for the authors: can they describe and comment the surgical procedure for haematoma evacuation?
Were there also other bleeding complications with the patients?
Did the patients have any other predisposing factors for bleedings? Please comment in the discussion or in the case presentations.

Author Response

Thank you very much for analyzing our article and your thorough evaluation. We made following changes in the text according to Your directions:

  1. can they describe and comment the surgical procedure for haematoma evacuation?

Ad. 1 Regarding the issue of surgical treatment of hematomas, we would like to emphasize that none of our patients underwent surgical treatment of SMH.From the literature analysis, it appears that large SMHs can be surgically evacuated by incision and drainage of the hematoma or via surgical excision and drainage (21). Another method is embolization of the bleeding vessel (percutaneous transcatheter arterial embolization, PTAE) in the case of active bleeding diagnosed by computed tomography (CT) (22).- the sentence was put into the text 2.      Were there also other bleeding complications with the patients? Did the patients have any other predisposing factors for bleedings? Please comment in the discussion or in the case presentations 

Ad. 2 . In the majority of our patients, there were no other risk factors for spontaneous bleeding other than anticoagulation or antithrombotic prophylaxis. One patient (case no. 2) had hypertension, which is listed as a risk factor for SMH. In addition, it can be assumed that intensive rehabilitation could lead to micro-injuries of the muscles. We found no other bleeding complications in any of the patients in this case series. All described patients had normal liver and kidney parameters, which were monitored several times during hospitalization.- the sentence was put into the text

Reviewer 2 Report

The authors reported seven cases of spontaneous intramuscular hematoma found in the lower limbs of spinal cord injury patients after antithrombotic prophylaxis or treatment was given. These patients did not have trauma when the hematoma was found. The number of SCI patients they included was 152, so the incidence of hematoma is 4.6%. 

In addition to the doses, the duration of these prophylaxes or treatments is also needed to be described in each case.  Did these patients have abnormal liver or kidney function?  What factors in these patients contributed to this complication? 

The manuscript was not well prepared, and it would be better if a native English speaker could edit it. 

Author Response

Thank you very much for analyzing our article and your thorough evaluation. We made following changes in the text according to Your directions:  

  1. In addition to the doses, the duration of these prophylaxes or treatments is also needed to be described in each case. 

Ad. 1 We have included the following table in response:

Table 2: A list of all antithrombotic treatment regimens implemented in a given case in chronological order untill the SMH formation.

Case number

Case 1

Case 2

Case 3

Case 4

Case 5

Case 6

Case 7

Type of therapy

1) Enoxaparin 0.4mg s.c -8 wk.

1)

Enoxaparin 0.4mg s.c -4 wk.

followed by

2) Enoxaparine 2 x 1mg/kg b.w. – 2 wk

1)

Enoxaparin 0.4mg s.c -9 wk,

followed by

2)

Enoxaparine 2 x 1mg/kg b.w. -3 wk;

3)

Dabigatran 2 x150mg 1 wk

1)

Enoxaparin 0.4mg s.c -9.4 wk,

1)

Enoxaparin 0.4mg s.c-4 wk

followed by

2)

Enoxaparine 2 x 1mg/kg b.w. -6 wk,

followed by

3)

Warfin 1.4 wk

1)

Enoxaparin 0.4mg s.c-7.7 wk

followed by

2)

rivaroxaban 20 mg -6.3 wk

1)

Enoxaparin 0.4mg s.c-10 wk

 Legend; wk- weeks, b.w.- body weight, s.c.-subcutaneus

  1. Did these patients have abnormal liver or kidney function?  What factors in these patients contributed to this complication? 

Ad. 2. In the majority of our patients, there were no other risk factors for spontaneous bleeding other than anticoagulation or antithrombotic prophylaxis. One patient (case no. 2) had hypertension, which is listed as a risk factor for SMH. In addition, it can be assumed that intensive rehabilitation could lead to micro-injuries of the muscles. We found no other bleeding complications in any of the patients in this case series. All described patients had normal liver and kidney parameters, which were monitored several times during hospitalization.– the sentence was put into the text

  1. The manuscript was not well prepared, and it would be better if a native English speaker could edit it. 

Ad. 3

 The manuscript was modified by  professional native English speaker.

Round 2

Reviewer 2 Report

My concerns have been resolved. 

Author Response

Dear Reviewer, 

We are happy that your concerns have been resolved.